



# Version 2 Ozone Monitoring Instrument SO₂ Product (OMSO2 V2): New Anthropogenic SO₂ Vertical Column Density Dataset

Can Li[1,2], Nickolay A. Krotkov[2], Peter J. T. Leonard[3,2], Simon Carn[4], Joanna Joiner[2], Robert J. D. Spurr[5], Alexander Vasilkov[6]

[1]Earth System Science Interdisciplinary Center, University of Maryland, College Park, MD 20742, USA
[2]NASA Goddard Space Flight Center, Greenbelt, MD 20771, USA
[3]ADNET Systems, Inc., Lanham, MD 20706, USA
[4]Michigan Technological University, Houghton, MI 49931, USA
[5]RT Solutions Inc., Cambridge, MA 02138, USA
[6]Science Systems and Applications Inc., Lanham, MD 20706, USA

*Correspondence to*: Can Li (can.li@nasa.gov)

**Abstract.** The Ozone Monitoring Instrument (OMI) has been providing global observations of SO₂ pollution since 2004. Here we introduce the new anthropogenic SO₂ vertical column density (VCD) dataset in the version 2 OMI SO₂ product (OMSO2

V2). As with the previous version (OMSO2 V1.3), the new dataset is generated with an algorithm based on principal component analysis of OMI radiances, but features several updates. The most important among those is the use of expanded lookup tables and model *a priori* profiles to estimate SO₂ Jacobians for individual OMI pixels, in order to better characterize pixel-to-pixel variations in SO₂ sensitivity, including over snow and ice. Additionally, new data screening and spectral fitting schemes have been implemented to improve the quality of the spectral fit. As compared with the planetary boundary layer SO₂ dataset in OMSO2 V1.3, the new dataset has substantially better data quality, especially over areas that are relatively clean or

affected by the south Atlantic anomaly. The updated retrievals over snow/ice yield more realistic seasonal changes in SO₂ at high latitudes and offer enhanced sensitivity to sources during wintertime. An error analysis has been conducted to assess uncertainties in SO₂ VCDs from both the spectral fit and Jacobian calculations. The uncertainties from spectral fitting are reflected in SO₂ slant column densities (SCDs) and largely depend on the signal-to-noise ratio of the measured radiances, as implied by the generally smaller SCD uncertainties over clouds or for smaller solar zenith angles. The SCD uncertainties for

individual pixels are estimated to be ~0.15-0.3 DU (Dobson Units) between ~40°S and ~40°N and to be ~0.2-0.5 DU at higher latitudes. The uncertainties from the Jacobians are approximately ~50-100% over polluted areas, and primarily attributed to errors in SO₂ *a priori* profiles and cloud pressures, as well as the lack of explicit treatment for aerosols. Finally, the daily mean and median SCDs over the presumably SO₂-free equatorial East Pacific have increased by only ~0.0035 DU and ~0.003 DU respectively over the entire 15-year OMI record; while the standard deviation of SCDs has grown by only ~0.02 DU or ~10%.

Such remarkable long-term stability makes the new dataset particularly suitable for detecting regional changes in SO₂ pollution.



# 1 Introduction

Despite substantial overall downward trends in recent years (*e.g.*, Aas et al., 2019; Klimont et al., 2013), sulfur dioxide ($SO_2$) emitted from anthropogenic sources (*e.g.*, coal-fired power plants) continues to have significant impacts on the environment. $SO_2$ is toxic and a regulated criteria air pollutant in many countries (*e.g.*, U.S. Environmental Protection Agency, 2010). It is

also a precursor to secondary sulfate aerosols that contribute to smog and haze (*e.g.*, Huang et al., 2014), cause acid deposition (*e.g.*, Likens et al., 1996), and influence regional climate (*e.g.*, Chuang et al., 1997; Haywood and Boucher, 2000). Removal of $SO_2$ and other short-lived pollutants is expected to bring health benefits (Lelieveld et al., 2019), but may also lead to additional warming (Samset et al., 2018). The ability to monitor global and regional changes in $SO_2$ pollution is thus critical for predicting and mitigating both air pollution and climate change.

Since the 1990s, a series of hyperspectral satellite sensors that measure solar backscattered radiances in the ultraviolet (UV) spectral range of ~300-400 nm has provided global monitoring of anthropogenic $SO_2$ (*e.g.*, Eisinger and Burrows, 1998; Lee et al., 2008; Nowlan et al., 2011; Theys et al., 2017; Valks and Loyola, 2008; Yang et al., 2013; Zhang et al., 2017). Among these sensors, the Ozone Monitoring Instrument (OMI) aboard the NASA Earth Observing System Aura spacecraft (Levelt et al., 2006) is particularly useful for $SO_2$ observations, thanks to its high spatial resolution (best at its launch in 2004) and daily

contiguous global coverage. The 15 year and growing OMI $SO_2$ data record is the longest among those from similar UV backscatter instruments (Levelt et al., 2018), having facilitated a number of studies on regional trends of $SO_2$ pollution. For example, OMI data have provided observational evidence on the efficacy of $SO_2$ control measures in China (Li et al., 2010; 2017a), confirmed significant further reductions in $SO_2$ emissions from the U.S (Fioletov et al., 2013; He et al., 2016) and Europe (Krotkov et al., 2016), and detected large recent increases in $SO_2$ pollution over India (*e.g.*, Li et al., 2017a; Lu et al.,

2013). OMI $SO_2$ data have also helped to quantify emissions from different types of sources (*e.g.*, Carn et al., 2017; Fioletov et al., 2016; Kharol et al., 2020; Zhang et al., 2019), to identify sources that are missing or underestimated in bottom-up emission inventories (McLinden et al., 2016), and to build a hybrid $SO_2$ inventory that combines both top-down and bottom-up emission estimates (Liu et al., 2018). More recently, OMI $SO_2$ data have been used to study changes in acid deposition over the eastern U.S. (Fedkin et al., 2019) and China (Zhang et al., 2020).

Several different techniques have been applied to OMI $SO_2$ retrievals. The first generation OMI standard $SO_2$ total vertical column density (VCD) product (OMSO2 V1.1 and earlier versions) is based on the band residual difference (BRD) algorithm (Krotkov et al., 2006) for planetary boundary layer (PBL) $SO_2$ VCDs (primarily for monitoring anthropogenic pollution), and the linear fit (LF) algorithm (Yang et al., 2007) for volcanic $SO_2$ VCDs. Both are discrete wavelength algorithms that only use a small subset of OMI wavelengths in the spectral range of interest. They are fast and sensitive to sources such as large power

plants and degassing volcanoes, but are relatively noisy and are prone to artifacts. Starting from OMSO2 V1.2, a new retrieval technique based on principal component analysis (PCA) of OMI-measured radiances (Li et al., 2013) was introduced to produce the OMI PBL $SO_2$ dataset. The PCA-based spectral fitting algorithm makes use of all available OMI wavelengths between 310.5 and 340 nm, suppressing retrieval noise by a factor of two as compared with the BRD algorithm and largely





eliminating unphysical biases over clean background areas. These improvements allow SO₂ point sources as small as 30 kt
($10^3$ t)/year to be quantified (Fioletov et al., 2015). In OMSO2 V1.3, an extended version of the OMI PCA algorithm (Li et al., 2017b) was developed for the updated volcanic SO₂ dataset. The same algorithm has also been implemented with the Ozone Mapping Profiler Suite (OMPS) Nadir Mapper aboard the NASA/NOAA Suomi National Polar orbiting Partnership (SNPP) spacecraft, thereby generating consistent retrievals with OMI (Li et al., 2017b; Zhang et al., 2017). Additionally, comparisons of OMI SO₂ retrievals between the PCA algorithm and a differential optical absorption spectroscopy (DOAS) algorithm (Theys
et al., 2015) also show generally good agreement.

While the OMI PBL SO₂ dataset produced with the PCA algorithm has significantly improved data quality as compared with the earlier version based on the BRD algorithm, the two algorithms share a common limitation: they both use a constant air mass factor (AMF) or SO₂ Jacobian spectrum for all pixels. The AMFs (or Jacobians) represent the sensitivity of OMI radiances to SO₂ total VCDs and they depend on several factors including ozone amount and profile, SO₂ *a priori* profile,
surface reflectivity, cloudiness, surface and cloud pressure, and solar and viewing zenith angles. The constant Jacobian spectrum used in the latest OMI PBL SO₂ dataset (OMSO2 V1.3) is pre-computed with the VLIDORT radiative transfer (RT) code (Spurr, 2008), assuming cloud-free conditions with SO₂ predominantly in the lowest 1 km of the atmosphere. The spectrum does not take into account variations in geometry, O₃, clouds, or surface reflectivity (*cf.* Li et al., 2013 for details). This simplification enhances computation efficiency, but also results in relatively large biases particularly for pixels over
cloudy scenes or background areas, and pixels near the edges of the swath where absorption due to O₃ can substantially change SO₂ Jacobians.

Here, we describe the version 2 OMI SO₂ total vertical column density product (OMSO2 V2). As with the previous versions, OMSO2 V2 includes datasets for both volcanic and anthropogenic SO₂. The volcanic SO₂ dataset in OMSO2 V2 is largely unchanged from OMSO2 V1.3. The anthropogenic SO₂ algorithm, on the other hand, has seen some major updates and will
be the focus of this paper. In particular, a set of new lookup tables and model-based *a priori* profiles are now used to estimate Jacobians for anthropogenic SO₂ retrievals for each individual pixel, thus better characterizing the sensitivity to SO₂ at different parts of the OMI sensor swath (*e.g.*, nadir pixels *vs.* swath edges), over different regions (*e.g.*, polluted *vs.* clean), and in different seasons (*e.g.*, summer *vs.* winter). This helps to further improve the retrieval quality. The rest of the paper is organized as follows: in Sect. 2 we provide a description of the new OMI anthropogenic SO₂ algorithm. This is followed by data quality
assessment in Sect. 3 and examples from the new anthropogenic SO₂ dataset in Sect. 4.

## 2 Algorithm Description

### 2.1 Algorithm overview

The new anthropogenic SO₂ algorithm for OMSO2 V2 introduces several new components but retains the overall framework of the original PCA-based OMI PBL SO₂ algorithm that has been described in detail elsewhere (Li et al., 2013). Briefly, the
algorithm employs a PCA technique to the measured radiance spectra from a number of satellite pixels to extract spectral



features in the form of principal components (PCs). The PCs are ranked in a descending order according to their spectral variance content. In the absence of large $SO_2$ plumes, the leading PCs that contain the most variance (see Figure 1 in Li et al. 2013 for an example) are typically associated with physical processes other than $SO_2$ absorption (*e.g.*, ozone absorption, rotational Raman scattering) or some measurement features (*e.g.*, wavelength shift, dark current). By fitting a set of $n_v$ PCs ($v_i$)

along with $SO_2$ Jacobians ($\frac{\partial N}{\partial \Omega_{SO_2}}$) to the measured radiances, we obtain an estimate of the $SO_2$ VCD ($\Omega_{SO2}$), at the same time minimizing the interferences from those processes represented by the PCs:

$$N\left(\omega, \Omega_{SO_2}\right) = \sum_{i=1}^{n_v} \omega_i v_i + \Omega_{SO_2} \frac{\partial N}{\partial \Omega_{SO_2}}, \qquad (1)$$

where $N$ is for the sun-normalized radiance spectrum for a satellite pixel in N-value ($N(\lambda) = -100 \times \log_{10}(I(\lambda)/F(\lambda)$, $I$ and $F$ are the measured radiance and solar irradiance at wavelength $\lambda$, respectively), and $\omega_i$ is the derived coefficient for the PC $v_i$. For

OMI retrievals, the algorithm processes one orbital swath at a time, with each swath comprising 60 rows across the flight direction of the Aura spacecraft (cross-track) and each row containing ~1600 pixels along the flight direction (along-track). For each swath, the algorithm also conducts PCA and spectral fit for each row separately, effectively treating them as different detectors.

The algorithm flowchart in Fig. 1 provides an overview of the new OMI anthropogenic $SO_2$ algorithm. It consists of three

main components: 1) preprocessing and data filtering in order to select certain pixels within an OMI row for PCA analysis; 2) initial estimates of $SO_2$ vertical column densities (VCDs) made assuming a constant Jacobian spectrum; and 3) determination of pixel-specific Jacobians and final estimates of $SO_2$ VCDs. More detailed descriptions of these components are given below.

## 2.2 Preprocessing and data filtering

An important prerequisite for the spectral fit in Eq. (1) to work properly is that the PCs contain minimal spectral structures

from $SO_2$ absorption. This condition is satisfied for the vast majority of atmospheric scenarios, given that background $SO_2$ loading is normally quite small (< 0.1 Dobson Units, 1 DU = $2.69 \cdot 10^{16}$ molecules/cm$^2$) over most areas. On the other hand, $SO_2$ light absorption can be substantial in the presence of large volcanic plumes or over heavily polluted areas, leading to apparent $SO_2$ structures in some of the leading PCs. The prepocessing and data filtering component of the algorithm aims to flag those pixels with strong $SO_2$ signals and exclude them from the PCA analysis, thus minimizing their impacts on the

spectral fit. Likewise, pixels having large solar zenith angles (SZA > 75°) or affected by the OMI row anomaly (signal suppression at certain OMI rows, see http://www.knmi.nl/omi/research/product/rowanomaly-background.php for more information) are also filtered out using the dynamic row anomaly flag from the OMI L1B data. It should be pointed out that $SO_2$ VCD retrievals are still attempted for all pixels with SZA < 75° and unaffected by the row anomaly, regardless of whether they are flagged for $SO_2$.

In the first step of data preprocessing, pixels with relatively large volcanic signals are flagged based on ozone retrieval residuals from two wavelength pairs (313/314 nm and 314/315 nm). We first estimate OMI radiances at these wavelengths using the





total column $O_3$ from the OMTO3 product (Bhartia, 2005) in conjunction with the simple Lambertian equivalent reflectivity (SLER) derived at the surface (Ahmad et al., 2004), assuming no $SO_2$. The residuals are the differences between the measured and estimated logarithmic radiances. When there is indeed little $SO_2$ in the atmosphere, the residuals are similar for the wavelength pairs (*e.g.*, 313 and 314 nm). For volcanic eruptions, however, the residuals at 313 nm become much greater due to stronger $SO_2$ absorption that is unaccounted for in $O_3$ retrievals. A large difference in $O_3$ residuals between 313 and 314 nm or between 314 and 315 nm for a given pixel thus signals relatively large abundance of $SO_2$, and that pixel is flagged. Side-by-side comparisons between the flagged pixels and OMI volcanic $SO_2$ retrievals (Li et al., 2017b) indicate that the flagging scheme is effective at identifying pixels with ~5 DU or more of $SO_2$ (assuming plume centered at ~18 km).

The second step of the preprocessing attempts to further screen for $SO_2$. After rejecting pixels with large SZAs or flagged for volcanic $SO_2$, the measured radiance spectra between 310.5 and 340 nm from the remaining (typically ~1200-1300) pixels in the row are subject to a PCA analysis. As large volcanic $SO_2$ signals have already been screened out, the first five derived PCs are usually free from $SO_2$ spectral structures. We fit those PCs to the measured radiances and calculate fitting residuals for each pixel. Without $SO_2$ structures in the PCs or the $SO_2$ Jacobian term on the right hand side (RHS) of Eq. (1), the fitting

residuals for a pixel having sizable $SO_2$ (*e.g.*, those over heavily polluted areas) are expected to be correlated with the $SO_2$ cross sections. We flag pixels that have a relatively large absolute cross product between the fitting residuals and a normalized spectrum of $SO_2$ cross sections.

In the final step of the preprocessing, a second PCA analysis is conducted for the radiance spectra within 310.5-340 nm from a given OMI row, this time excluding all pixels that have been flagged for $SO_2$. The resulted PCs are used as input to the

second component of the algorithm for initial estimates of $SO_2$ VCDs.

## 2.3 Initial estimates of $SO_2$ VCDs

To make initial estimates of $SO_2$ VCDs ($\Omega_{SO2\_ini}$), we carry out a spectral fit following Eq. (1) using the first 6 PCs from the final preprocessing step (see Sect. 2.2) along with a fixed $SO_2$ Jacobian spectrum identical to that for the PBL $SO_2$ retrievals in OMSO2 V1.3. Pixels that are not flagged for $SO_2$ in the preprocessing and have relatively small initial $SO_2$ VCDs, with

$-2\sigma < \Omega_{SO2\_ini} < 1.5\sigma$ (where $\sigma$ is the standard deviation of $\Omega_{SO2\_ini}$), are selected for a new round of PCA analysis. The updated PCs are then used in another spectral fit to produce updated VCD estimates (Fig. 1). Note that the threshold for pixel selection was set at $\pm1.5\sigma$ in our previous PBL $SO_2$ algorithm (Li et al., 2013). Now the lower limit has been relaxed to reduce the minor negative biases over some areas in the previous product. Additionally, the threshold is further relaxed by 50%, to $-3\sigma < \Omega_{SO2\_ini} < 2.25\sigma$ for pixels with SZA > 60°, considering that the VCDs at larger solar zenith angles tend to be noisier due to lower

signal-to-noise ratio in the radiance data.

This process of spectral fit, pixel selection, updated PCA, and updated spectral fit as described above is repeated three times. For the last two iterations, the OMI row is divided into three subsectors based on the SZAs: a tropical subsector with small solar zenith angles ( SZA < SZA$_{min}$ + 0.4 × (75° - SZA$_{min}$), where SZA$_{min}$ is the minimal SZA of the row), and two extratropical





ones south and north of it. As discussed in Li et al. (2013), PCs derived from these subsectors more closely represent local

observation conditions and reduce retrieval noise and biases. For these two iterations, we also use up to $n_v = 30$ PCs in the spectral fit. The exact value of $n_v$ is determined by checking the correlation between the PCs and the $SO_2$ cross sections. If for example the $i^{th}$ PC has significant $SO_2$ spectral structures, only $n_v = i - 1$ PCs are used in the spectral fit (*cf.* Li et al., 2013 for more discussion on the number of PCs included in the fit). With effective $SO_2$ screening steps in data preprocessing (Sect. 2.2), $n_v = 30$ PCs are used in the fit most of the time. After these three iterations, the finalized PCs are transferred to the third

component of the algorithm.

### 2.4 Determination of SO₂ Jacobians and final estimates of SO₂ VCDs

In the third and final component of the algorithm, we use the finalized PCs (Sect. 2.3) and the $SO_2$ cross sections (in place of Jacobians in Eq. (1)) to estimate $SO_2$ slant column densities (SCDs). In addition, we also use pixel-specific Jacobian spectra for the final $SO_2$ VCDs. To this end, we employ a table lookup approach and model-based *a priori* profiles to estimate $SO_2$

Jacobians for individual OMI pixels (Sect. 2.4.1 & 2.4.2). Special consideration is required for pixels covered by snow or ice (Sect. 2.4.3), as well as those over areas affected by the south Atlantic anomaly (SAA, Sect. 2.4.4)

### 2.4.1 Jacobian lookup tables

The total column $SO_2$ Jacobians ($\frac{\partial ln(I)}{\partial \Omega_{SO2}}$) and AMFs ($\frac{\partial ln(I)}{\partial \tau_{SO2}}$) represent the sensitivity of the natural logarithm of top of atmosphere (TOA), sun-normalized radiances ($I$) to perturbations in $SO_2$ VCD ($\Omega_{SO2}$) and $SO_2$ optical thickness ($\tau_{SO2}$) in the

entire atmospheric column, respectively. The two derivatives are linked through the absorption cross sections of $SO_2$. They can be calculated from the vertically resolved layer Jacobians (or box AMFs) and *a priori* profile of $SO_2$, for example:

$$\frac{\partial ln(I)}{\partial \Omega_{SO2}} = \int_0^{TOA} m(z) \, n_{SO2}(z) dz \, , \qquad (2)$$

where $n_{SO2}(z)$ is the normalized *a priori* profile or shape factor that represents the fraction of $SO_2$ molecules in layer $z$ to the overall number of $SO_2$ molecules in the entire column. The layer Jacobians (sometimes also referred to as scattering weight),

$m(z)$, are defined as the sensitivity of $I$ to changes in $\Omega_{SO2}(z)$, the partial column $SO_2$ density within layer $z$:

$$m(z) = \frac{\partial ln(I)}{\partial \Omega_{SO2}(z)} \, . \qquad (3)$$

Backscattered TOA radiances $I$ and layer Jacobians $m(z)$ at a wavelength $\lambda$ depend on several factors including $O_3$ (both the total amount and vertical distribution), observation geometry, and the pressure and reflectivity of the underlying clouds or surfaces. Following the same parameterization approach as in Li et al. (2017b), the backscattered TOA radiances in a Rayleigh

atmosphere can be calculated with the following equation:

$$I = I_0(\theta_0, \theta) + I_1(\theta_0, \theta) \cos \phi + I_2(\theta_0, \theta) \cos 2\phi + \frac{R I_T(\theta_0, \theta)}{(1 - R S_b)} \, , \qquad (4)$$

where $\theta_0$, $\theta$, and $\phi$ stand for SZA, VZA, and relative azimuth angle (RAA), respectively. $I_0$, $I_1$, and $I_2$ are Fourier expansion coefficients in $\phi$; together these terms represent the atmospheric component of $I$. The fourth term on the RHS of Eq. (4)





represents the surface component of $I$, in which $RI_r$ is the TOA radiance reflected once by the underlying surface (that has a

Lambertian reflectivity of $R$) and transmitted through the atmosphere, $S_b$ is the fraction of the surface-reflected radiance that

is scattered back to the surface, and $(1 - RS_b)$ accounts for the multiple reflections between the surface and the atmosphere.

The layer Jacobians can then be parameterized differentiating Eq. (4):

$$m(z) = \left[\frac{\partial I_0(\theta_0,\theta)}{\partial \Omega_{so2}(z)} + \frac{\partial I_1(\theta_0,\theta)}{\partial \Omega_{so2}(z)}\cos\phi + \frac{\partial I_2(\theta_0,\theta)}{\partial \Omega_{so2}(z)}\cos 2\phi + \frac{R}{(1-RS_b)}\frac{\partial I_r(\theta_0,\theta)}{\partial \Omega_{so2}(z)} + \frac{R^2 I_r(\theta_0,\theta)}{(1-RS_b)^2}\frac{\partial S_b}{\partial \Omega_{so2}(z)}\right] \cdot I^{-1}. \tag{5}$$

With Eqs. (4) and (5), we can determine the layer Jacobians for any given $\phi$ and $R$ from multidimensional lookup tables that

contain $I_0$, $I_1$, $I_2$, $I_r$ and $S_b$ and their derivatives with respect to $\Omega_{SO2}(z)$ for different SZAs, VZAs, underlying surface/cloud

pressures, and $O_3$ amounts and profiles. To account for the effects of $O_3$ on layer Jacobians, we use a climatology of ozone

profiles that depends on the total ozone amount developed by Labow et al. (2015) from sonde and Microwave Limb Sounder

(MLS) measurements. The profile Jacobian elements in Eq. (5) may be calculated conveniently using the VLIDORT radiative

transfer model, which has the ability to generate any sets of analytically calculated Jacobians in a polarized multi-layer

atmosphere. For each of the 46 ozone climatology profiles, we ran the VLIDORT to build a Jacobian lookup table (LUT) with

dimensions of $6 \times 8 \times 8 \times 72 \times 801$. The first three dimensions ($6 \times 8 \times 8$) correspond to the different nodes in the lookup table

for the underlying surface/cloud pressures, SZAs, and VZAs (see Table 1 for details). The last two dimensions are necessary

for storing vertically (72 layers, 0.01-1013.25 hPa) and spectrally (305-345 nm at 0.05 nm resolution) resolved parameters for

Jacobians.

The effects of clouds on $SO_2$ Jacobians are accounted for with the independent pixel approximation (IPA) approach that is

commonly employed in UV/VIS trace gas retrievals (*e.g.*, Ahmad et al., 2004; Koelemeijer et al., 2001; Martin et al., 2002;

Seftor et al., 1994). For each OMI pixel, we use multidimensional interpolation to estimate layer Jacobians for both cloudy

and cloud-free parts of the pixel. For the cloudy part, the optical centroid cloud pressure (OCP) from the OMI Raman scattering

cloud product (OMCLDRR, Joiner and Vasilkov, 2006), retrieved at wavelengths near the $SO_2$ fitting window, is used and $R$

is taken to be 0.8. For the cloud-free part, the surface reflectivity and surface pressure assumed in OMCLDRR retrievals are

used. The cloudy ($m_{cld}(z)$) and cloud-free ($m_{clr}(z)$) layer Jacobians at layer $z$ are weighted with the cloud radiance fraction

(CRF) in the $SO_2$ fitting window:

$$m(z) = m_{cld}(z)CRF + m_{clr}(z)(1 - CRF), \tag{6}$$

$$CRF = f_c\frac{I_{cld}}{I_{meas}}, \tag{7}$$

where $I_{meas}$ and $I_{cld}$ are measured TOA radiances and estimated cloudy radiances, respectively, and $f_c$ is the effective cloud

fraction retrieved by the OMCLDRR algorithm. Following Eq. (2), the interpolated layer Jacobian profile is combined with

the *a priori* profile shape factor selected based on the latitude, longitude, and time (month of the year) of the OMI measurement

(see Sect. 2.4.2) to produce $SO_2$ column Jacobians between 305 and 340 nm at 0.05 nm resolution. The high-resolution

Jacobians are then convolved using the OMI slit function.




### 2.4.2 GEOS-5 *a priori* profiles

For *a priori* profile shape factors ($n_{SO2}(z)$), we use GEOS-5 (Goddard Earth Observing System, Version 5) global model simulations (72 vertical layers, 0.5° latitude by 0.667° longitude horizonal resolution) for the time period 2004-2014. The output from GEOS-5 is sampled at the OMI overpass time and normalized against the model simulated total SO₂ VCD within each grid cell to produce monthly shape factor profiles. For each month of the year, these shape factor profiles are then averaged over the entire simulation period to generate monthly climatology profiles for use as *a priori* in our SO₂ retrievals.

### 2.4.3 Retrievals for pixels covered by snow or ice

From Eq. (5), one would expect that highly reflective snow/ice covered surfaces could enhance the sensitivity of OMI to SO₂ particularly at lower altitudes. Retrievals over these areas in the previous OMI PBL SO₂ dataset (OMSO2 V1.3) are however biased high, owing to the use of a constant Jacobian spectrum. Here we use OMCLDRR product and snow/ice flag in the OMI L1B data to identify pixels that are cloud-free and covered by snow/ice, following an approach proposed by Vasilkov et al. (2010). For OMI pixels flagged for snow/ice in the L1B data (OML1BRGU), we compare the terrain pressure with the effective scene pressure retrieved by the OMCLDRR algorithm (Joiner and Vasilkov, 2006). If the difference between the two is within 50 hPa, the pixel is likely cloud-free. We then assume a cloud fraction of zero and use the simple Lambertian equivalent reflectivity (SLER) derived for that pixel in Jacobian calculations. On the other hand, if the difference is greater than 100 hPa, the pixel is likely cloudy and the cloud fraction is set to one in Jacobian calculations. For pixels having scene and terrain pressure differences between 50 and 100 hPa, unambiguous cloud detection is not possible. While we still assume cloud-free conditions in the Jacobian calculations for such pixels, they are flagged and should be excluded from data analysis due to greater uncertainty.

### 2.4.4 Retrieval noise suppression for areas affected by the south Atlantic anomaly

Polar-orbiting satellite sensors like OMI are often subject to greater fluxes of high-energy particles when flying over areas affected by the south Atlantic anomaly (SAA), leading to larger noise in trace gas retrievals. Following Richter et al. (2011), we have implemented a two-step spectral fit to suppress SO₂ retrieval noise over the SAA region. The scheme examines the spectral fitting residual at each wavelength between 310.5 and 340 nm for all OMI pixels within the region of 0-45°S, 100°W-5°E. If a pixel has wavelengths with relatively large fitting residuals (beyond ±0.2 *N*-value), these wavelengths are excluded in a second step spectral fit that produces the final SO₂ VCD for the pixel. As shown in the following sections, this two-step scheme effectively reduces retrieval noise over the SAA region.

## 3 Quality Assessment

Uncertainties in the retrieved SO₂ VCDs arise from both the spectral fit and the Jacobian calculation parts of the algorithm that are largely independent from each other. Uncertainties in the former can originate from measurement errors as well as the basis



functions (i.e., the PCs), and can be represented as uncertainties in $SO_2$ SCDs. On the other hand, uncertainties in the $SO_2$ Jacobians are primarily due to the various assumptions inherent in the radiative transfer model, and also to uncertainties in input parameters used in the RT calculations. In this section, we assess the uncertainties (Sect. 3.1) and long-term stability (Sect. 3.2) in SCDs, and also discuss various factors that contribute to the uncertainties in $SO_2$ Jacobians (Sect. 3.3).

**3.1 Uncertainties in slant column densities (SCDs)**

Two different methods have been used to estimate uncertainties in $SO_2$ SCDs: the first based on the fitting residuals with an approach similar to that for DOAS uncertainty estimates and the other based on a statistical analysis of $SO_2$ SCDs over the remote Pacific.

**3.1.1 SCD uncertainties estimated from fitting residuals**

We can express the basis functions on the RHS of Eq. (1) in terms of a matrix $\mathbf{A}$ that has dimensions of $K \times M$. The number
of columns in $\mathbf{A}$, $M = n_v + 1$, represents the $n_v$ PCs plus the $SO_2$ cross section spectrum included in the least squares fit, whereas the row dimension $K$ is the number of OMI wavelengths at which the PCs and $SO_2$ cross sections are specified. Following a common approach for estimating uncertainty in DOAS spectral fitting (*e.g.*, Zara et al., 2018), the uncertainty ($\varepsilon_j$) in the $j^{th}$ fitted parameter is the square root of the $j^{th}$ diagonal element of the covariance matrix:

$$\varepsilon_j = \sqrt{\chi^2 (A^T A)^{-1}_{jj}} \,, \tag{8}$$

where $\chi^2$ can be calculated from the fitting residuals ($r(\lambda_k)$, the difference between the measured and fitted $N$-values) at all wavelengths in the fitting window and the degree of freedom ($K - M$):

$$\chi^2 = \frac{1}{K-M} \sum_{k=1}^{k=K} r^2(\lambda_k) \,. \tag{9}$$

The estimated SCD uncertainties for four selected OMI swaths over the remote Pacific in different seasons in 2007 are given in Fig. 2i-l, along with the SCDs (Fig. 2a-d) and scene reflectivity at 354 nm (Fig. 2e-h). As can be seen from the plots, the
SCD uncertainties for most pixels are within the range of 0.05-0.25 DU and demonstrate substantial spatial variability. To the first order, there is an apparent connection between the estimated uncertainties and the reflectivity. Pixels over bright surfaces covered by clouds (*e.g.*, east of New Zealand in orbit 13160) or snow/ice (*e.g.*, over the Antarctica) often have smaller uncertainties. This is probably due to enhanced signal-to-noise ratio in OMI measured radiances over highly reflective scenes, suggesting that the measurement noise is probably a driving factor for SCD uncertainties. The estimated SCD uncertainties
are also generally greater at higher latitudes, again probably reflecting strong light extinction and reduced signal-to-noise ratio at larger solar zenith angles and larger $O_3$ amounts. There is also a gradient in SCD uncertainties, for example, just south of Hawaii in orbit 13160. Recall that pixels from each row are grouped into three subsectors for the final PCA (*cf.* Sect. 2.4), based on their solar zenith angles. And the gradient is likely caused by the changes in basis functions (i.*e.*, the PCs) over the transition zones between the tropical and the extratropical subsectors. Additionally, the SCD uncertainties also show some



cross-track dependence particularly at lower latitudes, being mostly smaller on the eastern side of the swath than on the western side. The reason for this cross-track difference is not yet fully understood, but it could be due to stronger reflection of sky light on the eastern side of the swath over ocean (Vasilkov et al., 2017). Similar cross-track dependence has also been found over land and could be due to generally stronger scattering in the back scattering direction (on the eastern side) than in the forward scattering direction (on the western side), as demonstrated by Qin et al. (2019).

As expected, the retrieved SCDs are quite small over the remote Pacific covered by these swaths (Fig. 2a-d). As a result, the relative uncertainties (calculated as the ratio between the estimated uncertainties and the retrieved SCDs) are fairly large (see Fig. S1 in the supplemental information). About half of the pixels have relative uncertainties within ±100% and ~80% have relative uncertainties within ±275%. In contrast, pixels that have substantial real $SO_2$ signals (*e.g.,* downwind of the Kilauea volcano in Hawaii) have relative uncertainties of ~20-50%.

### 3.1.2 SCD uncertainties estimated from statistical analysis

Another common way to assess SCD uncertainties is to calculate the standard deviation of SCDs over background areas that have small natural variability in $SO_2$ such as the equatorial Pacific (*e.g.*, Li et al., 2013). In Fig. 2m-2p, we map the standard deviation of $SO_2$ SCDs within 2° latitude segments of each row for the same OMI swaths as in Fig. 2i-2l. Calculations are limited to 60°S-60°N, as SZAs and ozone amounts tend to be more variable at higher latitudes. Also, pixels with large SCDs
(> 1 DU) are excluded. As compared with the SCD uncertainties estimated from the fitting residuals (hereafter referred to as $\varepsilon_{SCD}$), the standard deviation of SCDs (hereafter referred to as $\sigma_{SCD}$) is considerably greater especially at higher latitudes. This is to be expected, given that $\sigma_{SCD}$ includes not only just noise from the spectral fit, but the natural variability in SCDs. For instance, $\sigma_{SCD}$ is enhanced downwind of Hawaii (Fig. 2p), likely reflecting larger variability caused by the $SO_2$ plume from the Kilauea volcano (Fig. 2d). As for the spatial pattern, there are similarities between $\sigma_{SCD}$ and $\varepsilon_{SCD}$, with both being generally
smaller at low latitudes and over clouds. As with $\varepsilon_{SCD}$, $\sigma_{SCD}$ also appears to be smaller on the eastern side of the swath.

A more detailed comparison between $\varepsilon_{SCD}$ and $\sigma_{SCD}$ for selected OMI rows can be found in Fig. 3. $\varepsilon_{SCD}$ is ~0.15 DU over the equatorial Pacific (20°S-20°N) and with few exceptions, is mostly < 0.2 DU at all latitudes. $\sigma_{SCD}$ shows much larger variability, ranging from ~0.2 DU near the equator to over 0.5 DU at around 60°S and 60°N in some cases. The difference between $\varepsilon_{SCD}$ and $\sigma_{SCD}$ is generally less than 0.1 DU at low latitudes but can exceed 0.2 DU at high latitudes. If we consider $\varepsilon_{SCD}$ as a lower
bound for SCD uncertainties and $\sigma_{SCD}$ as an upper bound, we arrive at the conclusion that the SCD uncertainties from the new algorithm are ~0.15-0.3 DU between ~40°S and ~40°N, and ~0.2-0.5 DU at higher latitudes. For a moderately polluted area in the middle latitudes (*e.g.*, ~30-40°N) with an SCD of ~0.3 DU, this translates into a relative uncertainty of ~50-100%.

### 3.2 Long-term changes in SCDs over remote background areas

Drift in retrievals over background areas may introduce artificial trends or mask actual trends over polluted regions, and the
long-term stability in the OMI anthropogenic $SO_2$ dataset is of great importance for detecting regional changes. Here we





examine temporal changes in SCDs from the new algorithm over the equatorial East Pacific (20°S-20°N,130-150°W) throughout the OMI record from October 2004 to December 2019. To ensure consistency in sampling, we only use data from rows 6-24 (1-based) that are considered to be minimally affected by the OMI row anomaly. Following McLinden et al. (2016), we also try to minimize the impact of volcanic eruptions. This is achieved by excluding days when the 99[th] percentile of $SO_2$

SCDs over the East Pacific (80°S-80°N,130-150°W) exceeds 0.8 DU. The daily median and mean of $SO_2$ SCDs for the equatorial East Pacific are shown in Fig. 4a and 4b, respectively. A linear regression analysis indicates that both median and mean SCDs have statistically significant (the linear correlation coefficient, $r = 0.18$, and $p < 0.05$ from two-tailed t-test) but very small changes over time, at $2.0 \cdot 10^{-4}$ and $2.3 \cdot 10^{-4}$ DU/year, respectively. In other words, over the entire ~15 year OMI record to date, the daily mean of background $SO_2$ SCDs has only increased slightly by ~ 0.0035 DU (~0.003 DU for median).

This long-term stability in the $SO_2$ record is an indicator of the stable performance of the OMI instrument itself (Schenkeveld et al., 2017), and it also confirms the ability of the PCA-based retrieval method to account for some of the drifts in measurements.

On the other hand, the standard deviation of $SO_2$ SCDs (Fig. 4c) over the equatorial East Pacific shows more notable changes, growing from ~0.19 DU in 2005 to ~0.21 DU in 2019 at a rate of ~$1.5 \cdot 10^{-3}$ DU/year. This represents an approximately 10%

increase in SCD uncertainties in ~15 years. The well below 1% per year rise in retrieval noise can be attributed to instrument degradation over time (Schenkeveld et al., 2017). A recent study (Zara et al., 2018) reports a faster growth rate in the uncertainties for OMI $NO_2$ retrievals (1-2% per year) and a more comparable rate for HCHO retrievals (< 1% per year). Note that $NO_2$ retrievals rely on measurements at visible wavelengths from a different detector (VIS) of the OMI instrument than $SO_2$ and HCHO retrievals (UV-2).

An examination of the daily 5[th] and 95[th] percentiles of the SCDs (Fig. 4d and 4e) over the same area reveals larger changes in opposite directions, at $-2.0 \cdot 10^{-3}$ and $2.7 \cdot 10^{-3}$ DU/year, respectively. As a result, the spread or the difference (Fig. 4f) between these two time series has increased by a total of almost 0.1 DU over the 15-year period. These changes in the percentiles suggest that the growth in the standard deviation is likely driven by more outliers in the retrieved SCDs. Indeed, the distribution of $SO_2$ SCDs over the equatorial East Pacific has grown broader since 2005 (see Fig. S2 for plots of probability density function

for different years). For 2005, 43.2% (98.5%) of the OMI pixels over the area have SCDs between -0.1 and 0.1 DU (-0.5 to 0.5 DU). The percentage has decreased to 41.3% (98.1%) by 2012, and further to 39.6% (97.4%) by 2019. Overall, the increase in the noise in the SCD retrievals is quite modest, pointing to good long-term stability of the new OMI anthropogenic $SO_2$ dataset.

### 3.3 Discussion on uncertainties in $SO_2$ Jacobians

In addition to the spectral fit, uncertainties in the $SO_2$ VCDs also depend on the Jacobian calculations. We have conducted several sensitivity tests using the VLIDORT RT code to investigate potential sources of uncertainties in Jacobians/AMFs. Note that these tests are not meant to be inclusive; rather, the aim is to shed some light on the relevance of different aspects in the

error budget. Detailed results of these tests can be found in the supplemental information (Figs. S3-S8). Here we summarize the potential error sources.

**1) Uncertainties in forward radiative transfer model assumptions** (such as $SO_2$ cross sections) and the table lookup interpolation scheme. Laboratory-measured cross sections usually have uncertainties < 10%, and RT codes have uncertainties of about 5% (Theys et al., 2017). As for LUT interpolation, our tests indicate that the associated uncertainties should be generally within 5-10% at altitudes relevant to anthropogenic $SO_2$ retrievals (Fig. S3).

**2) Uncertainties in *a priori* profiles**. Comparisons of monthly *a priori* profiles (Sect. 2.4.2) with available aircraft
measurements (*e.g.*, Dickerson et al., 2007) suggest that the climatology lacks the day-to-day variations associated with synoptic weather systems, and that may lead to ~15-40% of error for individual pixels over polluted regions such as northeastern China (Fig. S4). Model-simulated daily *a priori* profiles may better capture short-term changes in $SO_2$ vertical distribution but are currently not yet implemented in our retrievals.

**3) Uncertainties in surface reflectivity, cloud fraction, and cloud pressure**. Assuming a surface reflectivity of ~0.05, a
typical uncertainty of 0.01 causes ~7% uncertainty in $SO_2$ Jacobians under cloud-free conditions (Fig. S5). Depending on the vertical distribution of $SO_2$ and cloud height, clouds can either enhance (albedo effect) or reduce (shielding effect) OMI sensitivity. For polluted areas where $SO_2$ is predominantly in the lower troposphere, an uncertainty of ~0.05-0.1 in cloud fraction leads to an uncertainty of ~5-10% in Jacobians (Fig. S6), while an uncertainty of ~50-100 hPa in cloud pressure translates into ~25-40% uncertainty in Jacobians (Fig. S7).

**4) Lack of explicit consideration for aerosol effects on $SO_2$ Jacobians.** In the IPA approach (Sect. 2.4.1), the contribution of aerosols to TOA radiances is treated as if they arose from clouds. As a result, the aerosol scattering effects are implicitly accounted for by including aerosols as part of the effective cloud fraction. For non-absorbing or weakly absorbing aerosols (Fig. S8a and S8b), such implicit treatment may cause ~10-30% uncertainties in $SO_2$ Jacobians, but the sign (shielding *vs.* albedo effects) and the size of the errors are determined by the vertical distributions of aerosols and $SO_2$, as well as the
cloud/scene pressures from the cloud algorithm. For UV absorbing aerosols such as dust and smoke (Mok et al., 2016), the uncertainties can amount to ~50% (Fig. S8c).

The estimated uncertainties for the above aspects are mostly comparable with the error analysis conducted by Theys et al. (2017) for the TROPOMI $SO_2$ algorithm. Both studies point to the importance of $SO_2$ *a priori* profiles, cloud pressure, and the implicit treatment of aerosol effects, with the latter two associated with the IPA approach. If we assume that all these
uncertainty terms are independent, the overall uncertainties of column Jacobians/AMFs are estimated to be ~30-80% for individual pixels. For a polluted area where the uncertainties in SCDs are ~50-100%, the uncertainties in the retrieved $SO_2$ VCDs would be ~60-130% on an individual pixel basis. This is slightly higher than a previous estimate (45-110%) for SNPP/OMPS HCHO retrievals also using the PCA-based method (Li et al., 2015), as the present study considers additional error arising from the aerosol effects.



## 4 Example Results

In this section, we present examples from the new OMI anthropogenic SO$_2$ dataset (OMSO2 V2), focusing on the retrievals over snow/ice (Sect. 4.2) and long-term changes in SO$_2$ pollution over China and India (Sect. 4.3).

### 4.1 Comparison with the previous version OMI PBL SO$_2$ dataset

As an example, Fig. 5 compares the global mean SO$_2$ VCDs for May 2007 between the previous PBL SO$_2$ retrievals (Fig. 5a, OMSO2 V1.3), the new anthropogenic SO$_2$ retrievals using the same constant Jacobian spectrum (Fig. 5b), and the new retrievals using pixel-specific Jacobians (Fig. 5c). With the constant Jacobians, the differences between Fig. 5a and Fig. 5b are solely driven by changes in data screening and spectral fitting schemes (see Sect. 2.2 and 2.3). Overall, the two retrievals are quite similar. The most obvious difference is found over the areas affected by the SAA, where the mean (standard deviation) is 0.21 (0.48) and 0.08 (0.32) DU, respectively, for the previous PBL (Fig. 5a) and new (Fig. 5b) SO$_2$ VCDs. Outside of the SAA-affected areas, the two datasets are highly correlated, with a spatial correlation coefficient ($r$) of 0.89 and a slope of 0.9 (slope is ~1 is when using the reduced major axis method in the regression analysis). Over the equatorial Pacific, the new SO$_2$ VCDs in Fig. 5b also have a slightly more positive background than the PBL SO$_2$ retrievals (mean VCD: 0.06 *vs.* -0.03 DU), but the noise level is quite comparable (standard deviation: 0.11 *vs.* 0.10 DU). The pixel-specific Jacobians and GEOS-5 *a priori* profiles implemented for the retrievals in Fig. 5c further reduce noise and biases over background areas. For example, over the equatorial Pacific, the mean and standard deviation of SO$_2$ VCDs are 0.01 and 0.02 DU, respectively. The large reduction in SO$_2$ VCDs over northern Russia in Fig. 5c is due to revised Jacobian calculations over snow/ice and is further discussed in Sect. 4.2.

### 4.2 Retrievals over snow/ice

In Fig. 6, we examine the SO$_2$ VCDs over Norilsk, Russia, home to the world's largest anthropogenic SO$_2$ source (Norilsk Nickel smelters, Fioletov et al., 2016). As shown in Fig. 6c and 6d, there is a large seasonal change in SO$_2$ VCDs from the previous OMI PBL SO$_2$ dataset, likely caused by snow/ice effects in April that were previously unaccounted for. The maximal SO$_2$ VCD within the domain for April 2007 is 7.0 DU, a factor of two greater than the maximum of 3.5 DU for July of the same year. Similarly, the mean SO$_2$ VCD for April (0.14 DU) is over three times greater than that for July (0.04 DU). In comparison, the seasonal change in the new OMI anthropogenic SO$_2$ dataset over the same area is much smaller (Fig. 6a and 6b). The maximal SO$_2$ VCDs for the same two months are 2.5 and 3.4 DU, respectively, whereas the corresponding mean SO$_2$ VCDs are 0.07 and 0.04 DU. Note that the maximal and mean SO$_2$ VCDs for July are nearly identical between the two datasets, implying that the updated retrievals over snow/ice are at least partly responsible for the more gradual and realistic seasonal change in the new dataset.

For another case of retrievals over snow/ice, refer to Fig. 7 that compares the new OMI anthropogenic SO$_2$ VCDs over snow-covered and snow-free surfaces during a historic snowstorm in eastern China in January to February, 2008. Retrievals for snow



pixels (Fig. 7a) and snow-free pixels (Fig. 7b) are spatially well-correlated ($r = 0.68$), but retrievals over snow show generally stronger SO$_2$ signals over source areas that can be identified from the bottom-up emission inventory (Crippa et al., 2018) in Fig. 7c. This appears to provide evidence that highly reflective snow/ice surfaces can enhance OMI sensitivity to emission sources even at relatively large SZAs during wintertime, although only modest improvement in the correlation between OMI

VCDs and the bottom-up emissions is found here ($r = 0.29$ between bottom-up emissions and over snow VCDs *vs.* $r = 0.26$ between emissions and snow-free VCDs).

### 4.3 Long-term regional trends in SO$_2$ VCDs

With stable retrievals over background areas (Sect. 3.2), the new OMI anthropogenic SO$_2$ dataset is particularly suitable for monitoring long-term regional trends in SO$_2$ pollution. Here we examine the changes in OMI SO$_2$ VCDs over two polluted

regions, eastern China and India, from 2005 to 2019. To ensure consistent sampling throughout the entire period, we use data from the same set of OMI rows (6-24, 1-based) that are minimally affected by the row anomaly. We also focus on the warm season (April to October) when OMI has overall better sensitivity to SO$_2$. The results are presented in Fig. 8 for eastern China and in Fig. 9 for India. As shown in Fig. 8, SO$_2$ VCDs over China are much greater at the beginning of the OMI record than in recent years. The average total SO$_2$ mass within the domain, calculated by summing up the SO$_2$ mass from all grid cells that

have VCD > 0.1 DU each year, reached its peak at 27.5 kt in 2007 and then saw decreases in three consecutive years to the level of 15.6 kt in 2010, reflecting the effects of pollution control measures (Li et al., 2010) as well as the global financial crisis (Krotkov et al., 2016). The SO$_2$ mass rebounded to 20.0 kt in 2011 and remained relatively stable at around 16 kt, before starting to decrease sharply since 2014 to reach 5.7 kt in 2016 and 3.2 kt in 2019, marking a drastic drop of ~88% from the peak in 2007. As discussed in our previous study (Li et al., 2017a), such a large reduction in SO$_2$ over China is likely a result

of major efforts undertaken by the Chinese government to address air quality issues. Adjusting the threshold of VCD (*e.g.*, to 0.05 DU) leads to different estimates of the total mass (*e.g.*, to 29.6, 18.3, 8.8, 7.2 kt in 2007, 2010, 2016 and 2019, respectively), but does not significantly alter the overall relative trend.

For India, the trajectory of SO$_2$ pollution is quite different from that of China. The total SO$_2$ mass within the domain in Fig. 9 started at ~1.2 kt in the first few years, increasing to 3.7 kt in 2016 and remaining at approximately at the same level for the

next few years (*e.g.*, 3.6 kt for 2019). Again, adjusting the VCD threshold from 0.1 to 0.05 DU changes the absolute amount of the estimated SO$_2$ mass (to ~4 kt in 2005 and 2006, and 7.4 and 6.7 kt in 2016 and 2019, respectively), but the qualitative trend remains unchanged. The analysis here extends our previous study (Li et al., 2017a) and confirms the projection that India is indeed becoming the largest emitter of anthropogenic SO$_2$ in the world.

### 5 Conclusions

We have made extensive updates to the PCA-based OMI anthropogenic SO$_2$ retrieval algorithm for the version 2 OMI SO$_2$ product (OMSO2 V2). The most important change involves the use of expanded lookup tables and model-based *a priori*



profiles to account for the effects of various factors on $SO_2$ Jacobians for individual pixels, including observation geometry, ozone column amount and profile, cloud fraction and cloud pressure, surface pressure and reflectivity, as well as the vertical distribution of $SO_2$. Special consideration has also been given to retrievals over snow/ice. Other significant updates include

new schemes that screen for pixels having relatively large volcanic or anthropogenic $SO_2$ signals to minimize their impacts on the PCA analysis and also an updated spectral fitting scheme that suppresses noise over the areas affected by the SAA.

Both spectral fitting and Jacobian calculations contribute to the uncertainties in the new OMI anthropogenic $SO_2$ retrievals. The contribution from the spectral fit part is represented by uncertainties in SCDs, which are largely driven by the signal-to-noise ratio of radiance measurements. This is evidenced by generally lower SCD uncertainties over clouds and snow/ice-

covered surfaces and also at smaller SZAs. The SCD uncertainties on an individual pixel basis, estimated through both the fitting residuals and a statistical analysis, are ~0.15-0.3 DU between ~40°S and ~40°N and ~0.2-0.5 DU at higher latitudes. As for uncertainties in Jacobian calculations, the main contributions come from *a priori* profiles, cloud pressure and the lack of explicit treatment for aerosol effects. The overall uncertainties in Jacobians are estimated at ~50-100% over polluted areas. For a mid-latitude pixel with an $SO_2$ SCD of ~0.3 DU, typical of a moderately polluted area, the overall uncertainty in the

VCD is ~60-130%.

The long-term stability of OMI anthropogenic $SO_2$ retrievals has also been assessed by examining the daily statistics of SCDs over the equatorial East Pacific. The mean and median $SO_2$ SCDs show little change throughout the 15-year data record from 2004 to 2019, increasing by ~0.0035 DU and ~0.003 DU, respectively. This highlights the remarkable stability of both OMI measurements and the PCA-based retrieval approach that intrinsically accounts for some of the instrument drifts. The standard

deviation of SCDs, as a measure of retrieval noise, has increased by ~0.02 DU or ~10% since the beginning of the OMI mission, likely driven by more outliers in retrievals as suggested by the widening range between the 5[th] and 95[th] percentiles. Nonetheless, the annual increase in retrieval noise is well below 1% and comparable with or slower than the growth of noise in OMI HCHO and $NO_2$ retrievals (Zara et al., 2018).

Comparisons with the previous OMI PBL $SO_2$ dataset in OMSO2 V1.3 show that the new algorithm leads to further

improvements in data quality. When using the same Jacobians, the noise in VCDs over the equatorial Pacific is comparable between the two versions, but the updated spectral fit in V2 reduces the standard deviation in the monthly averaged VCDs over the SAA areas by about a third. The use of pixel-specific Jacobians further reduces retrieval noise over the background areas. Updated retrievals over snow/ice yield more gradual and realistic seasonal changes in $SO_2$ VCDs over the large source in Norilsk, Russia. Retrievals for snow-covered pixels over eastern China during a historic snowstorm in early 2008 also show

enhanced sensitivity to $SO_2$ sources, as compared with retrievals for snow-free pixels from the same period. Finally, $SO_2$ VCDs from the new anthropogenic $SO_2$ dataset show a continued reduction in $SO_2$ over eastern China since 2016 and a gradual overall increase over India from the beginning of the OMI record, confirming previous reports on the different trajectories of $SO_2$ pollution between the two countries.

In summary, the new OMI anthropogenic $SO_2$ dataset in OMSO2 V2 has several improvements over the previous PCA-based

OMI PBL $SO_2$ dataset in OMSO2 V1.3. Looking forward, we are planning additional updates to the next version OMSO2

product. These include the use of daily *a priori* profiles from model simulations that better capture day-to-day variations in $SO_2$ vertical distribution, explicit consideration of aerosol effects on $SO_2$ Jacobians, as well as a more comprehensive error analysis for Jacobian calculations to assign an estimated VCD uncertainty for each pixel.

**Code and data availability**

OMSO2 V2 data are publicly available, free of charge, at Goddard Earth Sciences Data and Information Services Center (https://disc.gsfc.nasa.gov/datasets/OMSO2_003/summary). Code used to analyze data and produce figures in this paper is available upon request from the corresponding author.

**Author contribution**

CL designed and implemented the retrieval algorithm, performed tests, and prepared the manuscript. NK, SC, JJ, and AV made
suggestions on the design of the retrieval algorithm. PL assisted in the production code for the retrieval algorithm. RS provided the VLIDORT RT code and assisted in the preparation of lookup tables. JJ first proposed PCA-based trace gas retrieval technique. JJ and AV provided OMCLDRR cloud data. All authors commented on the manuscript.

**Competing interests**

The authors declare that they have no conflict of interest.

**Acknowledgements**

We would like to thank the NASA Earth Science Division (ESD) Aura Science Team program for funding of the OMI SO2 product development and analysis (Grant # 80NSSC17K0240). The Ozone Monitoring Instrument (OMI) is a Dutch/Finnish instrument flying aboard the NASA Earth Observing System Aura spacecraft. The OMI project is managed by the Royal Meteorological Institute of the Netherlands (KNMI) and the Netherlands Space Agency (NSO).

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

**Tables**

Table 1. Nodes of the solar zenith angle (SZA), viewing zenith angle (VZA), and surface/cloud pressure, as used in the pre-computed $SO_2$ Jacobians lookup tables.

| Parameter | Nodes | | | | | | | |
|---|---|---|---|---|---|---|---|---|
| SZA | 0° | 15° | 30° | 45° | 60° | 70° | 77° | 81° |
| VZA | 0° | 15° | 30° | 45° | 60° | 70° | 75° | 80° |
| Surface/cloud pressure (hPa) | 243.2 | 374.9 | 526.9 | 638.3 | 841.0 | 1013.2 | | |




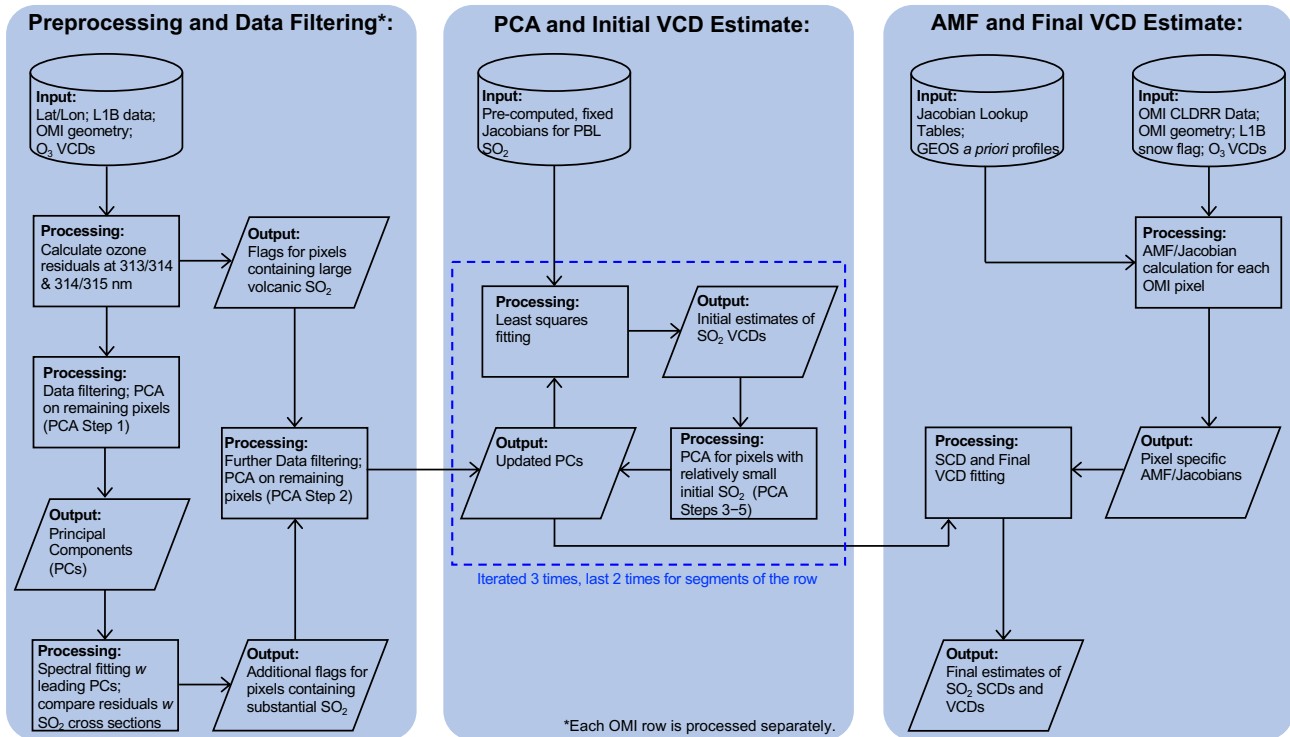

**Figure 1: Flowchart of the PCA-based spectral fitting algorithm for the anthropogenic SO₂ dataset in OMSO2 V2 product.**









**Figure 2: (a-d) The SO₂ slant column densities for four OMI swaths over the remote Pacific on (a) January 4, (b) April 3, (c) July 1, and (d) October 3, 2007. (e-h) The reflectivity at 354 nm for the same swaths. (i-l) Estimated uncertainties in SO₂ slant column densities ($\varepsilon_{SCD}$) for the same swaths. Cloudy or snow/ice covered areas have greater reflectivity and generally smaller errors in SO₂ SCDs. No retrievals were attempted for pixels with SZA > 75° that are grey-shaded. (m-p) The standard deviation of SO₂ SCDs ($\sigma_{SCD}$) within 2° latitude segments of individual OMI rows for the same swaths. The spatial coverage for (m-p) is limited to 60°S to 60°N as changes in observation conditions tend to be larger at higher latitudes. In addition, Pixels with SCD > 1 DU or SZA > 75°**
**and segments with < 10 valid pixels are excluded from the statistical analysis.**

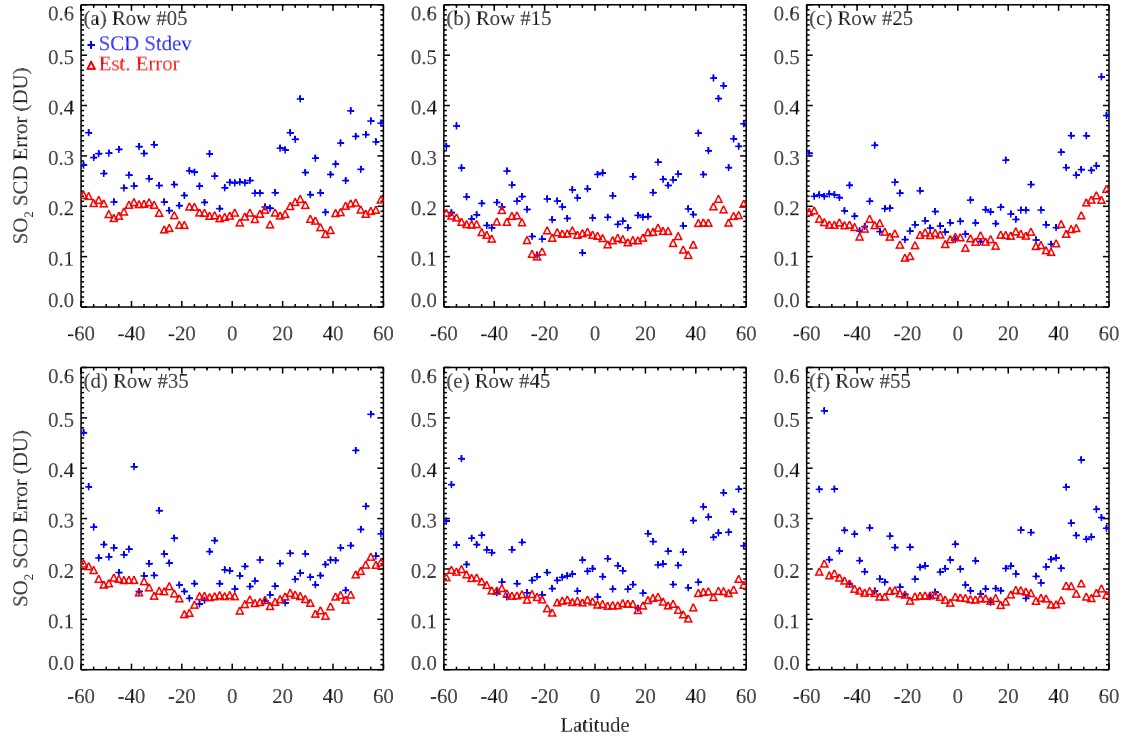

**Figure 3: Mean SO₂ SCD uncertainties estimated from fitting residuals ($\varepsilon_{SCD}$ , red triangles) and SCD standard deviation ($\sigma_{SCD}$, blue**
**plus signs) for 2° latitude segments between 60°S and 60°N from selected OMI rows from orbit 14456 on April 3, 2007.**

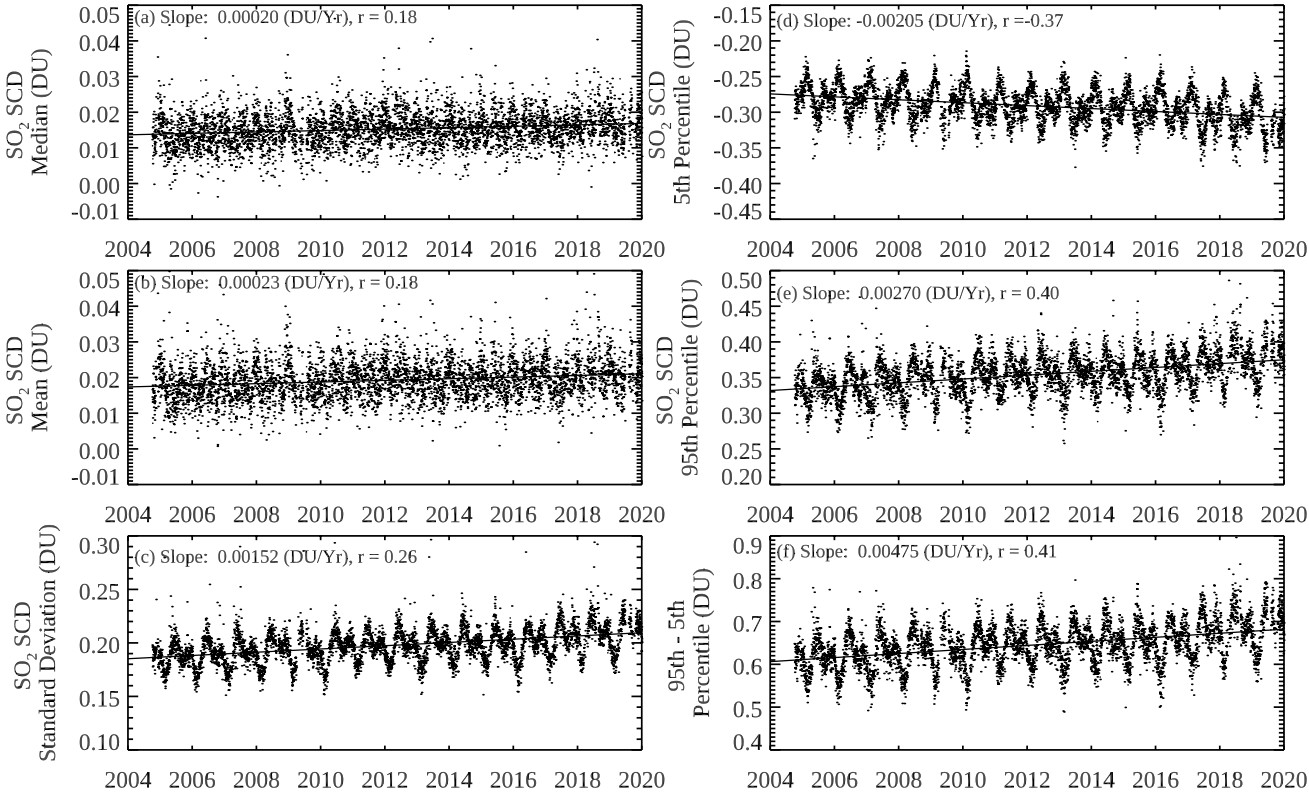

**Figure 4: Daily (a) median, (b) mean, (c) standard deviation, (d) 5th percentile, (e) 95th percentile and (f) the difference between 5th and 95th percentiles of OMI SO₂ SCDs over the equatorial East Pacific (20°S-20°N,130-150°W) during the period of 2004-2019 indicate stable long-term performance of the PCA-based anthropogenic SO₂ algorithm. Estimated linear trends and correlation coefficients are given for each time series. Only rows deemed to be minimally affected by the row anomaly are included in the statistical analysis. Days with possible influence of large volcanic SO₂ eruptions have also been excluded.**



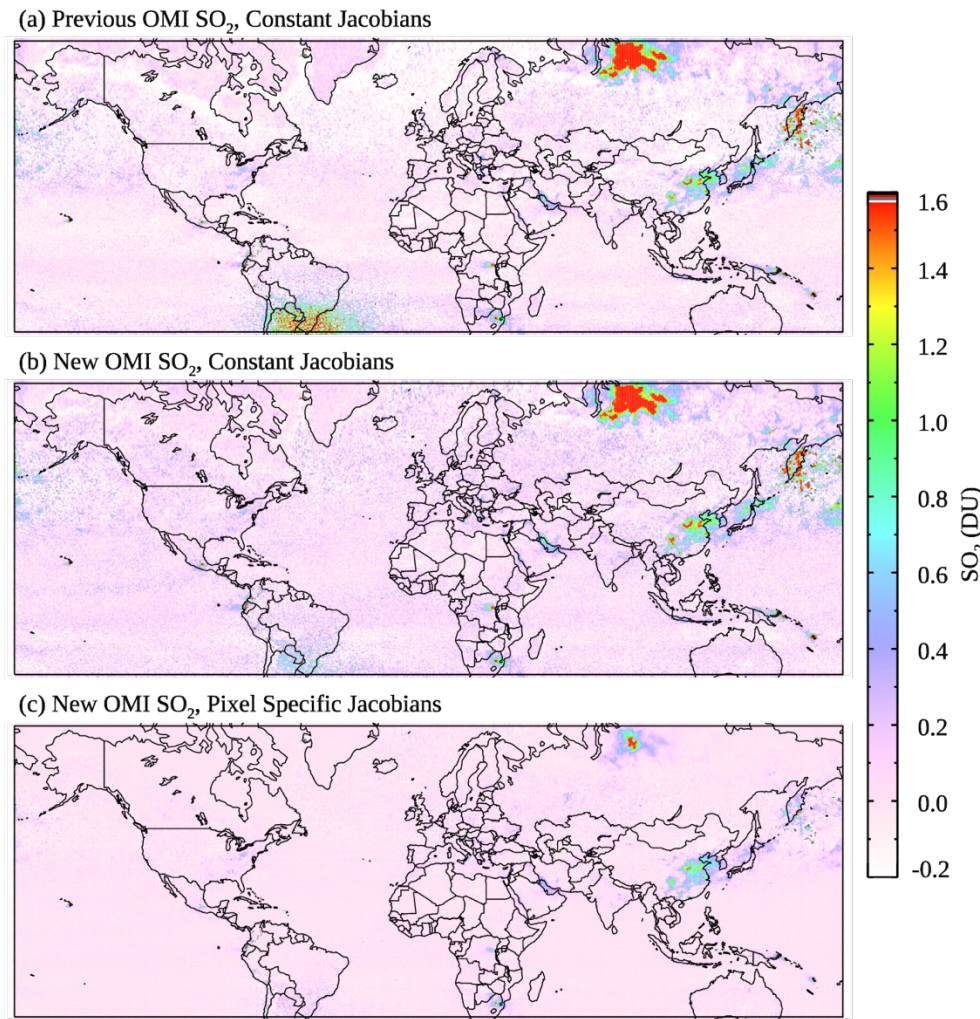

**Figure 5: Monthly mean OMI SO₂ VCDs for May 2007 from (a) the previous PBL SO₂ dataset in OMSO2 V1.3 using constant Jacobians, (b) the new OMI anthropogenic SO₂ retrievals using the same Jacobians as in (a), and (c) the new OMI anthropogenic SO₂ dataset in OMSO2 V2 using pixel-specific Jacobians as described in Sect. 2.4. Data are gridded to a horizontal resolution of 0.25° × 0.25°, and only those pixels near the center of the OMI swath (rows 6-55, 1-based), with SZA < 70°, and with a small cloud radiance fraction (CRF < 0.3) are included.**






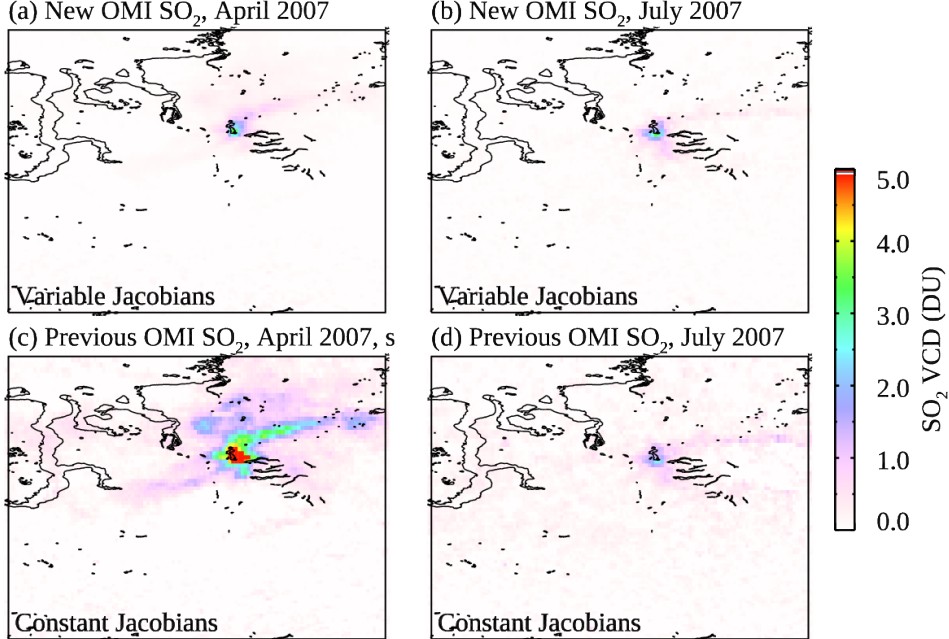

**Figure 6: Monthly mean SO₂ VCDs from the new anthropogenic SO₂ dataset in OMSO2 V2 for (a) April and (b) July, 2007 using pixel-specific Jacobians, showing comparable SO₂ over the area around Norilsk, Russia between the two months. Monthly mean SO₂ VCDs from the PBL SO₂ dataset in OMSO2 V1.3 show much greater apparent SO₂ in (c) April than in (d) July, due to the snow/ice effects that are unaccounted for with the constant Jacobians. Data have been filtered and gridded following the same criteria as in Fig. 5.**

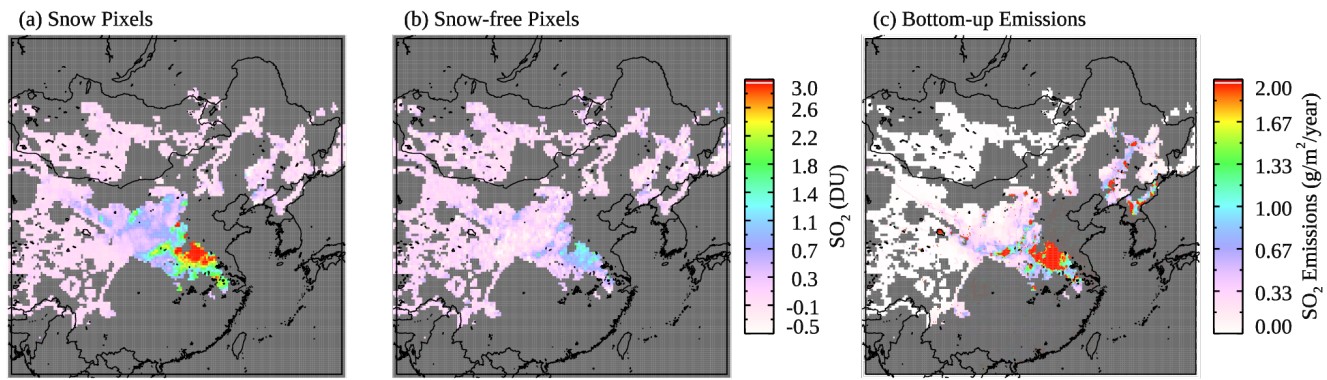

**Figure 7: Mean SO₂ VCDs from the new anthropogenic SO₂ dataset in OMSO2 V2 for January to February 2008 over eastern China from pixels identified to be (a) cloud-free and covered by snow and (b) cloud-free (CRF < 0.1) and snow-free. Retrievals over snow pixels have enhanced signals over source areas that can be identified from (c) bottom-up emission estimates. Only grid cells having at least 3 observations from both snow and snow-free pixels during the two month period are shown.**





**Figure 8: Mean warm season (April to October) OMI SO₂ VCDs over eastern China from the new anthropogenic SO₂ dataset in OMSO2 V2 for different years during 2005-2019. Data have been gridded to 0.25° × 0.25° resolution using pixels from OMI rows 6-24 (1-based) with cloud radiance fraction < 0.3, SZA < 65° and AMF at 313 nm > 0.3. Mean SO₂ VCDs are calculated from the daily gridded data for each year, after excluding days potentially affected by large volcanic plumes.**








**Figure 9: Same as Fig. 8 but for India.**