# Peer review of "Version 2 Ozone Monitoring Instrument SO2 Product (OMSO2 V2): New Anthropogenic SO2 Vertical Column Density Dataset"

_Atmospheric Measurement Techniques, 2020_

## Referee Comment (RC1) · Anonymous Referee #3 · 26 Jul 2020

1. For the abstract. Authors mentioned several versions of OMSO2. If there is a table in the content listing each versions' characterization that would be helpful? 2. Line 32. It might be better for readers to have a reference about how anthropogenic so2 emission "have significant impacts on the environment". 3. Line 73. Is it accurate to say the AMFs represent the sensitivity of radiances to SO2 VCD? Jacobians yes. But for AMF, is that just a factor between SCD and VCD? Although it can be affected by a lot of factors. 4. Line 83~84. Can I understand the new update in the version 2 is the improved anthropogenic SO2 product? 5. Line 117, "SO2 light absorption". Is this a precise expression? I guess you want to say the light (radiance) being absorbed by SO2. 6. Line 123, do you want to say SZA>75 degree? 7. Line 173,

what does sun-normalized radiances (I) mean? Is that the ratio of the back scattered radiance and the solar irradiance? But in line 181, the "I" has been defined as "Backscattered TOA radiance). 8. Line 242, Are you saying "within the SAA region of 0-45°, 100°W-5°E"? 9. Line 404 to 411. In this paragraph, are you comparing version 2 with snow/ice and without snow/ice? If yes, how do we evaluate the retrievals for the two case? By comparing the third party, you tell us, that SO2 retrieval over snow/ice covered surface might be more accurate than not be covered. How about the case of Norilsk in April and in July (starting from Line 394)? In that case, are you expressing that older version with constant Jacobian caused more seasonal change (snow cover and snow free)? 10. Line 416, when you say the OMI rows, do you mean the cross track number? 11. Why do you summing up the so2 mass for grid cells' VCD > 0.1? By what reason you selected two thresholds 0.5 and 0.1DU?

Please also note the supplement to this comment:
https://amt.copernicus.org/preprints/amt-2020-186/amt-2020-186-RC1-supplement.pdf

**Supplement:**

This paper provide an important documentation for a new global anthropogenic SO2 VCD dataset based on the OMI planetary boundary layer (PBL) SO2 product. It is useful for the air quality and environmental study. The paper described the principle component retrieval algorithm specific for the anthropogenic SO2 in the atmosphere. Conducted dataset quality assessment. Compared with several previous dataset versions. I would recommend that the paper be accept for publication. Following is my comments.

Comments

1. For the abstract. Authors mentioned several versions of OMSO2. If there is a table in the content listing each versions' characterization that would be helpful?
2. Line 32. It might be better for readers to have a reference about how anthropogenic so2 emission "have significant impacts on the environment".
3. Line 73. Is it accurate to say the AMFs represent the sensitivity of radiances to SO2 VCD? Jacobians yes. But for AMF, is that just a factor between SCD and VCD? Although it can be affected by a lot of factors.
4. Line 83~84. Can I understand the new update in the version 2 is the improved anthropogenic SO2 product?
5. Line 117, "SO2 light absorption". Is this a precise expression? I guess you want to say the light (radiance) being absorbed by SO2.
6. Line 123, do you want to say SZA>75 degree?
7. Line 173, what does sun-normalized radiances (I) mean? Is that the ratio of the back scattered radiance and the solar irradiance? But in line 181, the "I" has been defined as "Backscattered TOA radiance).
8. Line 242, Are you saying "within the **SAA** region of 0-45°, 100°W-5°E"?
9. Line 404 to 411. In this paragraph, are you comparing version 2 with snow/ice and without snow/ice? If yes, how do we evaluate the retrievals for the two case? By comparing the third party, you tell us, that SO2 retrieval over snow/ice covered surface might be more accurate than not be covered. How about the case of Norilsk in April and in July (starting from Line 394)? In that case, are you expressing that older version with constant Jacobian caused more seasonal change (snow cover and snow free)?
10. Line 416, when you say the OMI rows, do you mean the cross track number?
11. Why do you summing up the so2 mass for grid cells' VCD > 0.1? By what reason you selected two thresholds 0.5 and 0.1DU?

---

## Referee Comment (RC2) · Anonymous Referee #2 · 26 Aug 2020

The paper presents a new version of OMI SO2 algorithm and a dataset based on that algorithm. The new dataset has several improvements over the previous PCA-based OMI PBL SO2 dataset. It is an important step forward in the satellite SO2 retrievals. The paper is well-written and can be published after some minor revisions.

Comments:

The authors did not mention temperature dependence of SO2 absorption. It could play a role, for example, in the Norilsk case, when the different between winter and summer temperature could be as large as 50 degrees C.

P.6, l. 162. I wonder if the row anomaly had any impact on the number of selected PC.

[Figure]

What % of total variance is typically explained by them?

P. 10, l. 294 The condition that pixels with large SCDs (> 1 DU) are excluded could be too restrictive since the determined standard deviations were as high as 0.3 . What would happen if, for example, the limit was set to 2 DU?

p. 14, l. 410. It is not clear what these low correlation coefficients represent. The correlation coefficient depends on spatial resolution of the data as well as on the geographical region. From Figure 7, it looks that the correlation coefficient between a and c should be much higher if, for example, North Korea is excluded.

p.14, l. 417. This seems contradicts to the sentence above (l. 409) that says about winter snow/ice enchantment of SO2.

Figure 4, l. 664. Correlation coefficient with what? A linear function? How can we interpret these values? I think, you are trying to say something about statistical significance of the trend. Why do not you just give error bars for the slope?

---

## Author Response (AR1)

Response to review of "*Version 2 Ozone Monitoring Instrument SO$_2$ Product (OMSO2 V2): New Anthropogenic SO$_2$ Vertical Column Density Dataset*" (doi:10.5194/amt-2020-186)

Referees' comments in *Italic*, Responses in blue

**Anonymous Referee #3**
*This paper provide an important documentation for a new global anthropogenic SO2 VCD dataset based on the OMI planetary boundary layer (PBL) SO2 product. It is useful for the air quality and environmental study. The paper described the principle component retrieval algorithm specific for the anthropogenic SO2 in the atmosphere. Conducted dataset quality assessment. Compared with several previous dataset versions. I would recommend that the paper be accept for publication. Following is my comments.*

We thank the reviewer for the positive review and comments. We have revised our manuscript accordingly. Please find below our point-to-point response to specific comments.

*Comments*
*1. For the abstract. Authors mentioned several versions of OMSO2. If there is a table in the content listing each versions' characterization that would be helpful?*

Thank you for the suggestion. We have added a new table (Table 1) in the revised manuscript that summarizes the main features of different versions of OMSO2.

*2. Line 32. It might be better for readers to have a reference about how anthropogenic so2 emission "have significant impacts on the environment".*

To our knowledge, there has been a lack of comprehensive, recent reviews on the topic, and we have added a textbook (Seinfeld and Pandis, 2006) as a reference.

*3. Line 73. Is it accurate to say the AMFs represent the sensitivity of radiances to SO2 VCD? Jacobians yes. But for AMF, is that just a factor between SCD and VCD? Although it can be affected by a lot of factors.*

AMF (-dln(I)/dtau) and Jacobians (dln(I)/dSO2) are connected through SO$_2$ cross sections, and we can calculate one from the other. SCDs are determined by spectral structures in the radiance data, and they can be viewed as a quasi "measurement" quantity. Given the same VCDs (*i.e.*, identical amounts of SO$_2$), scenarios with larger AMFs (for example, due to more elevated SO$_2$ height) will have larger SCDs or equivalently more marked spectral structures in the measured radiances. Thus measurements will be more sensitive to SO$_2$ when the AMFs are larger, even if the absolute amounts (VCDs) are the same.

*4. Line 83~84. Can I understand the new update in the version 2 is the improved anthropogenic SO2 product?*

Yes, we have clarified this point in the revised manuscript and changed the sentence to "As compared with OMSO2 V1.3, the volcanic SO$_2$ dataset in OMSO2 V2 is largely unchanged,

while the anthropogenic $SO_2$ algorithm has seen some major updates and will be the focus of this paper."

*5. Line 117, "SO2 light absorption". Is this a precise expression? I guess you want to say the light (radiance) being absorbed by SO2.*

We have changed "$SO_2$ light absorption" to " the absorption of radiances by $SO_2$" in the revised manuscript.

*6. Line 123, do you want to say SZA>75 degree?*

We mean that retrievals are done for all pixels with SZA < 75 that are unaffected by the row anomaly, even if the pixels are flagged for $SO_2$ and excluded from PCA. We have clarified this in the revised manuscript.

*7. Line 173, what does sun-normalized radiances (I) mean? Is that the ratio of the back scattered radiance and the solar irradiance? But in line 181, the "I" has been defined as "Backscattered TOA radiance).*

Thank you for pointing this out. Yes, sun-normalized radiance $I$ is the ratio between the measured back scattered radiance at TOA ($I_{meas}$) and solar irradiance ($F$), *i.e.*, $I = I_{meas}/F$. In line 181 of the original manuscript, $I$ has the same definition. In the revised manuscript, we have made clarifications. We have also changed the symbol used following Eq. 1 to be more consistent with the rest of the paper.

*8. Line 242, Are you saying "within the SAA region of 0-45°, 100°W-5°E"?*

Yes. In the revised manuscript, we have changed the phrase to "within the SAA region (defined here as the domain 0-45°S, 100°W-5°E)".

*9. Line 404 to 411. In this paragraph, are you comparing version 2 with snow/ice and without snow/ice? If yes, how do we evaluate the retrievals for the two case? By comparing the third party, you tell us, that SO2 retrieval over snow/ice covered surface might be more accurate than not be covered. How about the case of Norilsk in April and in July (starting from Line 394)? In that case, are you expressing that older version with constant Jacobian caused more seasonal change (snow cover and snow free)?*

Yes, in line 404-411, we are comparing snow and snow-free retrievals in OMSO2 V2 over China. We have clarified this in the revised manuscript, adding " Both Fig. 7a and Fig. 7b show retrievals from OMSO2 V2, but Fig. 7a is for retrievals over snow-covered pixels and Fig. 7b is for those over snow-free pixels."

We agree that it is important to use third-party measurements to confirm that snow/ice retrievals in winter are indeed more sensitive to retrievals over snow-free areas. In the absence of such data, we have changed the last sentence in paragraph in the revised manuscript. The sentence now reads: "This appears to provide evidence that highly reflective snow/ice surfaces can

enhance OMI sensitivity to emission sources even at relatively large SZAs during wintertime, although a more thorough evaluation using other datasets such as those from ground monitors is necessary before a more definite conclusion can be drawn."

As for the Norilsk case, yes, the larger seasonal changes in OMSO2 V1.3 over the area are likely caused by snow/ice effects in April that are not accounted for with the constant Jacobians. In the revised manuscript, we have changed the second sentence of the paragraph to "The area is usually covered by snow/ice in April but not in July. As shown in Fig. 6c and 6d, there is a large seasonal change in $SO_2$ VCDs from the previous OMI PBL $SO_2$ dataset (OMSO2 V1.3), likely caused by snow/ice effects in April that were previously unaccounted for with the use of constant Jacobians."

*10. Line 416, when you say the OMI rows, do you mean the cross track number?*

Yes, in the revised manuscript, we have added that OMI rows are also referred to as cross-track positions at the first use of "OMI rows" (in section 2.1).

*11. Why do you summing up the so2 mass for grid cells' VCD > 0.1? By what reason you selected two thresholds 0.5 and 0.1DU?*

As $SO_2$ retrievals are still relatively noisy, for grid cells with VCDs below the threshold value, we assume that the actual $SO_2$ signal is too weak to be reliably detected by OMI, and thus we effectively treat them as if they contained no $SO_2$. This helps to reduce the effects of retrieval noise and small negative biases over certain areas.

To select the threshold value, we consider the typical retrieval noise for a PBL $SO_2$ profile (~0.5 DU, 1-sigma), and the number of measurements available for averaging each year (~50-100). Through data averaging, we can expect to reduce retrieval noise roughly by the square root of the number of measurements. So averaging 100 measurements would reduce the noise by roughly a factor of 10 (in an ideal case). As a result, we selected 0.1 DU as the threshold.

Since the selection of the VCD threshold is somewhat arbitrary, we also conducted sensitivity tests with different thresholds to see if the overall trends remain unchanged. We found that using a threshold of 0.05 DU did not qualitatively change our results.

We have added the above discussion to the revised manuscript.

Response to review of "*Version 2 Ozone Monitoring Instrument SO₂ Product (OMSO2 V2): New Anthropogenic SO₂ Vertical Column Density Dataset*" (doi:10.5194/amt-2020-186)

Referees' comments in *Italic*, Responses in blue

**Anonymous Referee #2**
*The paper presents a new version of OMI SO2 algorithm and a dataset based on that algorithm. The new dataset has several improvements over the previous PCA-based OMI PBL SO2 dataset. It is an important step forward in the satellite SO2 retrievals. The paper is well-written and can be published after some minor revisions.*

We thank the reviewer for the positive review and several good suggestions. We have made changes to the manuscript following the suggestions. Please find below our point-to-point response to specific comments.

*Comments:*
*The authors did not mention temperature dependence of SO₂ absorption. It could play a role, for example, in the Norilsk case, when the different between winter and summer temperature could be as large as 50 degrees C.*

Thank you for pointing this out. We have performed some calculations assuming different temperatures (243 K and 293 K below 2 km) and found that the temperature dependence of SO₂ cross sections could have contributed to the Norilsk case, although the effect is likely less than 10% (see figure below). We have added the results of these calculations in the updated supplemental information (Figure S3 in the revised version). We have also added some discussion to sections 3.3 and 4.2 of the revised manuscript.

[Figure]

SO₂ column Jacobians calculated assuming different temperatures (T) for the lowest 2 km of the atmosphere. All calculations assume SZA = 30°, VZA = 0°, RAA = 90°, middle latitude O₃ profile with $\Omega_{O3}$ = 325 DU, surface pressure = 1013.25 hPa, cloud fraction = 0, and SO₂ mostly below 1 km. The reference Jacobians (black) are used in OMSO2 V1.2 and 1.3 PBL SO₂ retrievals, assuming a temperature of 284 K at the surface, decreasing to 270 K at 2 km. At 310.8 nm, the differences in Jacobians (reference - test case) are 5.1% and -3.2% between the reference and test cases assuming constant temperatures of 243 K and 293 K below 2 km, respectively.

The differences at 313 nm are 1.6% and -0.9% between the reference and the 243 K and 293 K cases, respectively.

*P.6, l. 162. I wonder if the row anomaly had any impact on the number of selected PC. What % of total variance is typically explained by them?*

We exclude row anomaly affected pixels in the PCA and retrievals, so the row anomaly would have a minimal effect on the number of PCs, if any. We have clarified this in the revised manuscript. When we first developed the PCA-based retrieval technique, we did test the algorithm on row-anomaly affected pixels, and we found that we could not obtain retrievals with suitable quality. Thus the decision was made to exclude them from the algorithm.

*P. 10, l. 294 The condition that pixels with large SCDs (> 1 DU) are excluded could be too restrictive since the determined standard deviations were as high as 0.3 . What would happen if, for example, the limit was set to 2 DU?*

We have updated Figures 2 and 3 in the revised manuscript using a 2 DU threshold for SCDs instead of 1 DU. At lower latitudes, there are no discernable changes due to this replacement. At higher latitudes, we notice that the standard deviation increases for some (relatively few) segments. Overall, the use of different thresholds does not affect our general conclusions.

*p. 14, l. 410. It is not clear what these low correlation coefficients represent. The correlation coefficient depends on spatial resolution of the data as well as on the geographical region. From Figure 7, it looks that the correlation coefficient between a and c should be much higher if, for example, North Korea is excluded.*

We agree that the low correlation coefficient is not conclusive. We have removed the discussion related to this issue in the revised manuscript.

*p.14, l. 417. This seems contradicts to the sentence above (l. 409) that says about winter snow/ice enchantment of SO2.*

While the presence of snow/ice surfaces can enhance signals, for India and for most areas in China, the number of days with snow cover is actually rather low, and the sample size is generally too small for long-term data analysis. Warm season provides more retrievals that can be used to reduce noise through averaging. We have clarified this in the revised manuscript.

*Figure 4, l. 664. Correlation coefficient with what? A linear function? How can we interpret these values? I think, you are trying to say something about statistical significance of the trend. Why do not you just give error bars for the slope?*

Thank you for the suggestion. We have removed the correlation coefficient and added the 95% confidence interval for the slope in each panel of the figure.

[revised manuscript text omitted]

(a) Orbit 13160, 01/04/2007    (b) Orbit 14456, 04/03/2007    (c) Orbit 15752, 07/01/2007    (d) Orbit 17121, 10/03/2007

SO$_2$ SCD (DU)
-0.30  -0.20  -0.10   0.00   0.10   0.20   0.30

(e) Orbit 13160, 01/04/2007    (f) Orbit 14456, 04/03/2007    (g) Orbit 15752, 07/01/2007    (h) Orbit 17121, 10/03/2007

Scene Reflectivity 354 nm
0.00   0.10   0.20   0.30   0.40   0.50   0.60

(i) Orbit 13160, 01/04/2007    (j) Orbit 14456, 04/03/2007    (k) Orbit 15752, 07/01/2007    (l) Orbit 17121, 10/03/2007

SO$_2$ SCD Error (DU)
0.00   0.05   0.10   0.15   0.20   0.25   0.30

(m) Orbit 13160, 01/04/2007    (n) Orbit 14456, 04/03/2007    (o) Orbit 15752, 07/01/2007    (p) Orbit 17121, 10/03/2007

SO$_2$ SCD Stdev (DU)
0.00   0.05   0.10   0.15   0.20   0.25   0.30

[revised manuscript text omitted]

*Correspondence to*: Can Li (can.li@nasa.gov)

[Figure]

**Figure S1: The relative uncertainties in OMI SO₂ SCDs for the same four swaths as in Fig. 2, calculated as the ratio between the estimated uncertainty and the SCD for each pixel with SZA < 75°. The median relative uncertainty is ~30%, whereas the 5th, 10th, 25th, 75th, 90th and 95th percentiles are approximately -550%, -275%, -100%, 105%, 280% and 560%, respectively. Relative uncertainties for pixels with real SO₂ signals, for example, downwind of the Kilauea volcano in Hawaii are ~20-50%.**

[Figure]

**Figure S2: The probability density functions of OMI SO$_2$ SCDs over (a) the East Pacific and (b) the equatorial East Pacific for 2005 (black), 2012 (red) and 2019 (blue). The percentages of pixels that have SO$_2$ SCDs within the range of -0.1 to 0.1 DU and that of -0.5 to 0.5 DU for each year are also marked in the figure. Only pixels from rows 6-24 that are minimally affected by the row anomaly are included in the plots. Data from days potentially affected by large volcanic SO$_2$ plumes are also excluded.**

[Figure]

**Figure S3: SO$_2$ column Jacobians calculated assuming different temperatures (T) for the lowest 2 km of the atmosphere. All calculations assume SZA = 30°, VZA = 0°, RAA = 90°, middle-latitude O$_3$ profile with $\Omega_{O3}$ = 325 DU, surface pressure = 1013.25 hPa, cloud fraction = 0, and SO$_2$ mostly below 1 km. The reference Jacobians (black) are used in OMSO2 V1.2 and 1.3 PBL SO$_2$ retrievals, assuming a temperature of ~284 K at the surface, decreasing to 270 K at 2 km. At 310.8 nm, the differences in Jacobians (reference - test case) are ~5.1% and -3.2% between the reference and test cases assuming constant temperatures of 243 K and 293 K below 2 km, respectively. The differences at 313 nm are 1.6% and -0.9% between the reference and the 243 K and 293 K cases, respectively.**

[Figure]

Figure S4: Profiles of box AMFs from VLIDORT direct calculations (blue) and box AMFs interpolated from precomputed lookup tables (red) for different $O_3$ profiles, SZAs and VZAs. All four cases here have surface reflectivity R = 0.03 or 0.05. Moreover, interpolation errors for these cases, defined as the relative differences between the interpolated and the directly calculated box AMFs, are generally < 10% at all levels between 0 and 15 km altitude. Cases with higher reflectivity (not shown) tend to have smaller interpolation errors.

[Figure]

Figure S5: (a) Aircraft-measured $SO_2$ profiles over a polluted area in northeastern China in April 2005 show large reductions in $SO_2$ in just few days due to passage of a frontal system, leading to substantial changes in AMFs, calculated at 313 nm using the measured profiles assuming SZA = 30°, VZA = 0°, RAA = 90°, middle latitude $O_3$ and temperature profiles with $\Omega_{O3}$ = 325 DU, cloud fraction = 0, surface pressure = 1013.25 hPa and surface reflectivity R = 0.05. (b) AMF calculated using the monthly climatology *a priori* profile over the same area for April (black) is biased high and does not reflect the day-to-day variation in (a). AMFs calculated using daily model profiles (from MERRA-2, red for April 5, blue for April 7) better capture the effects of the changing $SO_2$ profiles on AMFs.

[Figure]

Figure S6: SO$_2$ column Jacobians calculated assuming different values of surface reflectivity (R) indicating that an error of 0.01 in R would lead to about ~7% error in AMFs at 313 nm under these particular conditions. All calculations assume SZA = 30°, VZA = 0°, RAA = 90°, middle latitude O$_3$ and temperature profiles with $\Omega_{O3}$ = 325 DU, surface pressure = 1013.25 hPa, cloud fraction = 0, and SO$_2$ mostly below 1 km.

[Figure]

Figure S7: Profiles of box AMFs calculated assuming different cloud radiance fractions (CRF), cloud pressure = 800 hPa, SZA = 30°, VZA = 0°, RAA = 90°, middle latitude O$_3$ and temperature profiles with $\Omega_{O3}$ = 325 DU, surface pressure = 1013.25 hPa, R = 0.05, and a PBL SO$_2$ profile (green line). An error of 0.05 in the cloud radiance fraction under these assumptions leads to ~5% error in AMF at 313 nm.

[Figure]

60

Figure S8: Profiles of box AMFs calculated assuming different cloud pressures, cloud radiance fraction = 0.5, and with the other conditions the same as those in Figure S6. Under these specifications, an error of 50 hPa in cloud pressure leads to up to ~25% error in AMF, while an error of 100 hPa leads to an error of almost 40%.

[Figure]

65  Figure S9: SO$_2$ Jacobians calculated with explicit corrections for the effects of (a) sulfate aerosols, (b) weakly absorbing industrial aerosols and (c) strongly absorbing smoke aerosols. For all types of aerosols, the same aerosol optical depth (AOD = 1.0 at 354 nm), size distribution (lognormal distribution with the effective radius, $r_{eff}$ = 0.1 μm) and vertical distribution (an exponential profile with the scale height, H = 0.9 km) are assumed, but the single scattering albedo is different (ϖ = 0.99, 0.95 and 0.83 at 354 nm, respectively). All other assumptions are the same as those in Figure S5 with R = 0.05. Also plotted in coloured lines are SO$_2$ Jacobians

70  estimated using the independent pixel approximation (IPA) approach to implicitly account for the aerosol effects. For the IPA estimates, an apparent effective cloud fraction is derived from the synthetic radiances with aerosols, assuming a cloud reflectivity of 0.8. A few different cloud/scene pressures are also assumed for scenarios with sulfate aerosols. The implicit aerosol correction in the IPA approach can lead to errors as large as 50% for smoke aerosols, with smaller errors for non-absorbing aerosols. Additionally, these errors have strong dependence on the vertical distributions of aerosols and SO$_2$.

---

## Author Response (AR2)

Dear Andreas,

Thank you for accepting our manuscript "Version 2 Ozone Monitoring Instrument $SO_2$ Product (OMSO2 V2): New Anthropogenic $SO_2$ Vertical Column Density Dataset" for publication in AMT. We appreciate your guidance through the peer review process.

We also appreciate your comment on the use of a threshold in $SO_2$ mass calculation. Per your suggestion, we have added a short comment on the biases this practice may lead to, in section 4.3 of the revised manuscript. We hope this correction has addressed your concern on the analysis.

Thank you again for a smooth peer-review process and for helping to improve the manuscript. Should you have any further comments or suggestions, please don't hesitate to let us know.

Best Regards,
Can Li

[revised manuscript text omitted]